# The Gut Microbiota May Affect Personality in Mongolian Gerbils

**DOI:** 10.3390/microorganisms10051054

**Published:** 2022-05-20

**Authors:** Lin Gan, Tingbei Bo, Wei Liu, Dehua Wang

**Affiliations:** 1State Key Laboratory of Integrated Management of Pest Insects and Rodents, Institute of Zoology, Chinese Academy of Sciences, Beijing 100101, China; ganlin@ioz.ac.cn (L.G.); botingbei@ioz.ac.cn (T.B.); 2University of Chinese Academy of Sciences, Beijing 100049, China; 3CAS Center for Excellence in Biotic Interactions, University of Chinese Academy of Sciences, Beijing 100049, China; 4School of Life Sciences, Shandong University, Qingdao 266237, China

**Keywords:** gut microbiome, animal personality, boldness, gut microbiota transplantation (FMT), *Meriones unguiculatus*

## Abstract

The “gut–microbiota–brain axis” reveals that gut microbiota plays a critical role in the orchestrating behavior of the host. However, the correlation between the host personalities and the gut microbiota is still rarely known. To investigate whether the gut microbiota of Mongolian gerbils (*Meriones unguiculatus*) differs between bold and shy personalities, we compared the gut microbiota of bold and shy gerbils, and then we transplanted the gut microbiota of bold and shy gerbils into middle group gerbils (individuals with less bold and shy personalities). We found a significant overall correlation between host boldness and gut microbiota. Even though there were no significant differences in alpha diversity and beta diversity of gut microbiota between bold and shy gerbils, the Firmicutes/Bacteroidetes phyla and *Odoribacter* and *Blautia* genus were higher in bold gerbils, and *Escherichia_shigella* genus was lower. Furthermore, the fecal microbiota transplantation showed that changes in gut microbiota could not evidently cause the increase or decrease in the gerbil’s boldness score, but it increased the part of boldness behaviors by gavaging the “bold fecal microbiota”. Overall, these data demonstrated that gut microbiota were significantly correlated with the personalities of the hosts, and alteration of microbiota could alter host boldness to a certain extent.

## 1. Introduction

Bacterial–host interactions are ubiquitous, and thus, the influences of bacteria on animals vary from subtle to profound. The “gut–microbiota–brain axis” has long been considered a network of connections involving multiple biological systems, allowing bidirectional communication between gut bacteria and the brain, and is crucial in maintaining homeostasis of the gastrointestinal and central nervous microbial systems of animals [1,2]. For example, the study of Brandt’s vole showed that the interaction between gut microbiota and norepinephrine (NE) regulates heat production through the vagus nerve [3]. In recent years, increasing evidence indicates that gut microbiota plays a crucial role in orchestrating the function of immune systems [4], metabolism [5] and the development of various organs [6]. In addition, there is other empirical evidence showing the gut microbiota can influence host behaviors through chemical communication (e.g., short-chain fatty acids) with the nervous system [1]. Previous studies suggest that the gut microbiota can affect host foraging behavior [7], anxiety-like behavior [8], activity [9] and social behavior [10]. A recent study found that the composition and diversity of gut microbiota are related to the personality characteristics of human infants [11]. However, we do not fully understand the relationship between personalities and gut microbiota in small mammals.

Animal personality has been defined as a consistent and relatively stable among-individual variation in behavior that is present in a wide variety of taxa [12]. In the past two decades, researchers have discovered that the animal personality is related to the immune system [13,14] and metabolism [9,15,16] states, which in turn affects the development of organs [17,18] and the variation of behaviors [19,20,21]. The “State–behavior feedbacks” theory regards individual state variations as the reason for individual behavior differences, and individuals thus differ in personalities because they are in a different state, adjusting their behaviors in an adaptive fashion to these differences [22]. Logically, the gut microbiota may affect animal personalities via the mechanism of the “gut–microbiota–brain axis”. For example, the gut microbiota may influence animal behavior through its metabolite ‘direct’ (e.g., short-chain fatty acids, SCFAs) [23] or ‘indirect’ (e.g., gut microorganisms are capable of synthesizing neurotransmitters themselves and can induce production of neurotransmitters by their animal hosts) [24] pathways to affect individual personalities.

Mongolian gerbils (*Meriones unguiculatus*) is a rodent species that lives in the steppe, semi-desert and desert habitats in northern China, southeast Mongolia, and the southern TransBaikal and south of Tuva region of Russia [25,26]. Mongolian gerbils live in social groups comprising 2–18 individuals throughout the year [27], where each group occupies an exclusive territory and all group members share the burrow system [28]. Previous studies have shown that Mongolian gerbils exhibit individual-specific trespassing by neighbors and chases involving individuals from adjacent groups are frequently observed in fields [28,29], and among-individual food-hoarding behaviors variation in the laboratory [29,30]. Recently, the gut microbiota regulating host physiological and behavioral processes has been reported in Mongolian gerbils [31]. However, individual variation of gut microbiota and its relationship with individual behavior variation in small mammals are still rarely reported. Here, we hypothesized that the composition of gut microbiota was correlated with the host’s boldness, and studied the relationship between boldness and gut microbiota, and specifically addressed two questions: (i) Is the boldness of gerbils related to the community structure of gut microbiota? (ii) If so, can gut microbiota transplantation change the bold character of gerbils?

## 2. Materials and Methods

### 2.1. Animals

All gerbils were offspring of Mongolian gerbils trapped in Inner Mongolian grasslands in 1999 and raised at the Institute of Zoology, the Chinese Academy of Sciences in Beijing [32]. Gerbils were housed in plastic cages (30 × 15 × 20 cm; 3–4 per cage) with sawdust bedding (3–5 cm) after weaning and were maintained at a room temperature of 23 ±  1 °C under a photoperiod of 16L:8D. Adult male gerbils (about 5 months old) had free access to water and food (standard rat pellets from Beijing Ke Ao Feed Co., Beijing, China). Gerbils were licensed under the Animal Care and Use Committee of the Institute of Zoology at the Chinese Academy of Sciences.

### 2.2. Experimental Procedures

We measured the boldness of 136 male Mongolian gerbils at 8 to 12 months of age. Additionally, in order to make our data more objective, we chose the boldest and shyest gerbils (*n* = 14, each) collecting their feces for gut microbiota measurement, and as donors for gut microbiota transplantation. Then, we selected 20 middle scores of gerbils into two experimental groups. One group, called “bold fecal” gavage (BG) group, received bold gerbils’ gut microbiota, and the other group, called “shy fecal” gavage (SG) group, received shy gerbils’ gut microbiota. Gerbils were housed single-caged and housed at 23 °C with free access to food and water. Then, gerbils were given antibiotics via intragastric gavage daily for 14 days. One day after the final gavage, the bacterial suspension (200 µL) was then transplanted via intragastric gavage to antibiotic-treated gerbils daily for 7 days [3,33]. After fecal microbiota transplantation, gerbils were housed separately for 7 days to eliminate the cage effect [34]. Finally, we measured the gerbil’s boldness again to observe the impact of gut microbiota on personality.

### 2.3. Ethical Statement

All experiments involving animals complied with the ASAB/ABS Guidelines for the Use of Animals in Research and were approved by the Institutional Animal Use and Care Committee of the Institute of Zoology, Chinese Academy of Sciences (Ethical Inspection License No: IOZ20190071). We strived to maximize the health of the animals and reduce their suffering.

### 2.4. Personality Assay

“Boldness” was defined as an individual’s reaction to any risky situation in a familiar situation [35]. Each gerbil’s boldness was measured by elevated plus-maze (EPM) during the light period [36,37]. We placed individual gerbils at the hub where the open and closed arms crossed and faced a closed arm, and recorded for 5 min. To determine the consistency and repeatability of boldness for gerbils, a second EPM test was run 1 week later. Entered or not, time spent and distance moved in the open arm, ratios of open arms entries to closed arms entries (ROE), ratios of open arms time spent to closed arms time spent (ROT) and ratios of open arms moving distance to closed arms moving distance (ROM) was used to create a boldness score. We used Pearson correlation coefficients to determine the correlation factors, and the related factors were used in the principal component analysis (PCA) to create composite behavioral scores [38,39].

### 2.5. Gut Microbiome Community Composition

Fresh feces were immediately frozen and stored at −80 °C after being collected from the excretion. DNA was extracted by 2× cetyltrimethyl ammonium bromide and phenol chloroform mixture (phenol: chloroform: isoamyl alcohol = 25:24:1), and was then isolated by a spin column using the SanPrep Column DNA Gel Extraction Kit (Sangon Biotec, 273 B518131-0100). Universal primers were used for PCR amplification of the V3-V4 hypervariable regions of 16S rRNA genes and contained Illumina 3′ adaptor sequences as well as a 12-bp barcode. Sequencing was pair-end on the Illumina HiSeq 2500 platform, with the sequencing strategy PE250. Raw sequencing reads were denoised, filtered according to barcode and primer sequences (Forward primer-341F, CCTACGGGNGGCWGCAG; Reverse primer-805R, GACTACHVGGGTATCTAATCC), and classified with the Quantitative Insights Into Microbial Ecology (QIIME2, version 2021.11.0) software suite, according to the Qiime2 tutorial (https://qiime2.org/, Accessed: 11 October 2021) with modified some methods [40]. Further filter out noisy sequences, error correction, remove chimeric sequences, remove singletons was performed using DADA2 [41] algorithm (“qiime dada2 denoise-paired”), and the remaining tags were clustered into amplicon sequence variants (ASVs). The taxonomy of these features was assigned to the Silva database (release 132), and a feature table was generated using Qiime2’s classify-sklearn taxonomy classifier. Alpha diversity of fecal microbiota was characterized by Chao1, Faith’s phylogenetic diversity and Shannon diversity indices using “qiime diversity alpha” command line. Statistical comparison of the alpha diversity indices between group levels was performed using the Wilcoxon rank-sum test. Principal coordinate analysis (PCoA) plots of the Bray–Curtis metric were calculated with square root transformed data and visualized in R (vegan package Version 2.5-4). Permutational multivariate analysis of variance (PERMANOVA) was performed to disclose the factors shaping the dynamics of the feces microbial communities between bold and shy gerbils. PERMANOVA, a distribution-free algorithm, accommodates random effects, repeated measures, and unbalanced datasets [42]. For PERMANOVA analysis, we used the adonis function in the vegan package of R including different independent variables (e.g., bold and shy gerbils) with default settings (Bray–Curtis distance and 999 permutations). Stage-dependent features were identified by using the linear discriminant analysis (LDA) effect size (LEfSe) with default settings (e.g., LDA score > 2) [43].

The function prediction of the microbial community was performed using PICRUSt2 [44] based on ASVs clustered from 16S rRNA sequencing data, and then metabolic predictions were identified from Kyoto Encyclopedia of Genes and Genomes (KEGG) pathway hierarchy levels 2 and 3 for interpretation and subsequent analysis [45]. The difference in predicted results was processed using the linear discriminant analysis (LDA) effect size (LEfSe) with default settings (e.g., LDA score > 2).

### 2.6. Fecal Microbiota Transplant (FMT)

Gerbils were treated with fresh composite antibiotics (containing 100 μg/mL neomycin, 50 μg/mL streptomycin, and 100 U/mL penicillin; Sigma, Darmstadt, Germany) via intragastric gavage (200 μL) daily for 14 days prior to the start of chemotherapy [3]. For fecal microbiota transplantation, 200 mg of the donor’s feces was diluted in 2 mL of 0.9% sodium chloride solution and centrifuged to obtain the bacterial suspension. The bacterial suspension (200 µL) was then transplanted via intragastric gavage to antibiotic treated gerbils daily for 7 days [3,33].

### 2.7. Statistical Analysis

We used a principal component analysis (package ‘stats’) to reduce the number of behavioral variables, and applied the Kaiser–Guttman criterion (eigenvalue > 1) [46] when selecting the number of components to retain [38,39]. The MCMC generalized linear mixed model (package ‘MCMCglmm’) is used to assess individual boldness consistency [47] with the number of experiments as a fixed effect and the gerbil’s ID as a random effect, and the 95% confidence intervals of the repeatability by running 1000 permutations of each test. We used Pearson correlation to analyze the association of six boldness behavior variables and used two-way ANOVA to test whether BG and SG differ in behavior variables and whether the behavior variables are different between fecal microbiota transplantation before and after (SPSS 20.0, SPSS Inc., Chicago, IL, USA). Variations in the beta diversity of bacterial communities were statistically compared using analysis of similarity (ANOSIM, permutations = 999). We also compared the relative abundances of microbial genera using the pipeline LEfSe (linear discriminant analysis (LDA) effect size), using an LDA score threshold of 3. The unweighted/weighted UniFrac distance metrics were compared with the distance matrix of the boldness score using Mantel’s test in order to test the correlations between the host bold and gut microbiota. Differences between groups were statistically analyzed using the Kruskal–Wallis test with a level of *p* < 0.05 (* *p* < 0.05, ** *p* < 0.01, *** *p* < 0.001). Results were presented as means ± SEM.

## 3. Results

### 3.1. Individual Boldness Behavior

The PCA reduced the numbers of exploratory variables to 2 components (Table 1), which, combined, explained 91.87% of the total variance. The first component (PC1) explained 84.04% of the variance, and the second component (PC2) explained 7.83% of the total variance. PC1 is the only component with an eigenvalue > 1 and negatively with all the behavior parameters in EPM. Thus, we used PC1 in our study since it explained the majority of the variance and was strongly correlated with behavior during the EPM, which is an index of boldness [36]. Then, we referred to the PC1 values as the boldness scores, and the scores of low to high meant bold to shy. The MCMCglmm on boldness score revealed the consistent differences in boldness score over time between individuals with repeatability of R = 0.555 (95% confidence interval 0.399–0.657).

### 3.2. Gut Microbiota

From the Rank Abundance, we found this curve decreases gently, indicating that the distribution of bacterial species is uniform, and the uniformity program of species is not different (Appendix A). Chao1 species richness (alpha diversity) was not different between the bold and shy gerbils (*p* = 0.064, Figure 1A). Shannon index and Faith’s phylogenetic diversity were also not significantly different between the two groups (*p* = 0.511 and *p* = 0.062, Appendix A). PCoA based on Bray–Curtis distance showed no significant difference between bold and shy gerbils (Permanova, *p* = 0.223, Figure 1B). PCoA based on unweighted and weighted unifrac distance were also showed no significant difference between bold and shy gerbils (Permanova, *p* = 0.483 and *p* = 0.252, Appendix A). We observed differences in amplicon sequence variant (ASVs) abundance at the phyla level and found Firmicutes and Bacteroidetes were the most abundant phyla in all gerbils, while Firmicutes and Bacteroidetes were a little lower in Shy gerbils (Figure 1C). The heat map showed that the microbial composition between the two groups was not well separated at the genus level (Figure 1D). We analyzed the influence of boldness on the microbiome at a finer scale by comparing the proportions of the top 30 most abundant OTUs. At the genus level, we found the *Odoribacter* and *Blautia* abundance were significantly higher in bold gerbils, while the *Escherichia_shigella* was significantly higher in shy gerbils (LefSe, LDA > 2, Figure 1E). To assess the metabolic potential of the intestinal microbiota, PICRUSt2-based functional prediction revealed differences in microbial functions in the intestinal microbial communities between bold and shy gerbils. (Figure 1F). We found porphyrin and chlorophyll metabolism were higher in bold gerbils, and taurine, hypotaurine and tryptophan metabolism were higher in shy gerbils.

Pearson correlation analysis of six boldness indicators showed that there was a very significant correlation between them (*p* < 0.01). The Mantel test between the genus abundance of gut microbiota and boldness score showed microbial abundance was closely linked to ratios of open to closed arms entries (ROE, *p* = 0.034) and moving distance in the open arms (*p* = 0.055) as revealed by the Mantel test (Figure 2).

### 3.3. Boldness Behavior Changes after FMT

The boldness score of the boldness assessment showed there was no difference between the BG and SG groups before and after gut microbiota transplantation, which indicated that change in gut microbiota did not affect boldness in Mongolian gerbils (Figure 3). We next analyzed the change in six behavioral parameters of boldness after FMT. We found that the time spent in the open arms, the moving distance in the open arms, ROT and ROM were increased but not significantly in BG gerbils, and other behavioral parameters remained unchanged after FMT (Figure 4).

## 4. Discussion

In this study, we found a significant correlation between host boldness and gut microbiota. Additionally, we found that the alpha and beta diversity of gut microbiota were not significant different between bold and shy gerbils, but the *Odoribacter* and *Blautia* abundance of bold gerbils was higher and *Escherichia_shigella* was lower than shy gerbils. We found that alterations in the gut microbiome did not significantly affect the boldness score of Mongolian gerbils in the FMT experiment, but “bold feces” can reinforce receptors’ bold behavior phenotypes. Therefore, our results suggest that the gut microbiota plays a role in the personality regulation of Mongolian gerbils.

Most of the “bold fecal” gavage (BG) gerbils enhanced 66.9% (95% confidence interval 20.1–113.6%) of their bold behaviors (time spent in the open arms, moving distance in the open arms, ROT and ROM) after FMT, while in the “shy fecal” gavage (SG) group, the effects were chaotic. This finding is unexpected since the state-behavior feedbacks hypothesis suggests that individual behavior differences are caused by individual state variations and that individuals adjust their behavior in an adaptive way to accommodate these differences [22]. Recent empirical studies have shown that the gut microbiota are involved in the regulation of the host physiological states, such as metabolisms and hormone levels [2,23,48]. Thus, we suggested that changes in gut microbiota would affect the host state and further alter the host personalities. However, the gut microbiota did not exhibit a significant difference between bold and shy gerbils, only some of the bacteria’s abundance differences. Furthermore, the results showed only BG gerbils changed boldness behaviors (increased but not significant) after fecal microbiota transplant.

One of the potential reasons to explain our results is that the function of gut microbiota may differ in different phyla or genus. Indeed, the unique function of specific microbiota phyla/genus has been found in many studies as follows: Proteobacteria is related to the biosynthesis of vitamins and contributes to the breakdown and ferment of complex sugars for the host [49]; Firmicutes contribute to the production of enzymes involved in fermenting vegetative material and have the potential to fabricate vitamin B [50]; *Anaerostipes* can protect against an allergic response to food [51]; *Odoribacter* has specific microbiota strains and immune mechanisms to enhance host immune [52]. Our results showed that the gut microbiota of gerbils is mainly Firmicutes and Bacteroidetes, while *Odoribacter* was higher in bold gerbils. We hypothesized that *Odoribacter* was the most important gut microbiota in determining boldness. Indeed, *Odoribacter* and *Blautia* are two known butyrate producers [53,54], which are involved in improving the host metabolism [54,55], and the butyrate was able to regulate systemic metabolism by passing through the enterocytes to influence the peripheral tissues [56,57]. Furthermore, the host’s high activity level has been reported to be positively associated with the abundance of the genus *Odoribacter* and *Blautia* [58,59], while high metabolism and activity are thought to be significant characteristics of the bold personality [60,61]. Accordingly, we proposed that *Odoribacter* and *Blautia* abundances are the decisive factors in determining the gerbil’s boldness.

The causal relationship between personality and gut microbiota might be an alternative explanation for our results, namely that personality shows a dominant position in the correlation to boldness and gut microbiota. The model of Sih et al. (2015) suggests the joint emergence and maintenance of among-individual differences in behavior and state, and how such differences are promoted by positive feedback between behavior and state. For example, bold individuals would have higher activity level or more aggressive, which required them to have a higher metabolic rate to support their behavior phenotypes, and bold individuals via their personality-dependent behavior and physiological characteristics to influence the gut microbiota composition. One study found that socially dominant rats could be distinguished from subordinates based on their intestinal microbiota: Clostridiaceae, Prevotellaceae, and Bifidobacteriaceae were significantly enriched in dominant rats, which were associated with butyrate production, whereas Veillonellaceae were less represented [62]. The FMT experiment showed that gut microbiota was the key to maintaining social dominance, and sodium butyrate could enhance the social dominance of mice [62]. Another study found that the gut microbiota composition of dogs was based on their body condition [63], which suggested that the state of the host may determine the composition of the gut microbiota. As a result, we hypothesized that personality-dependent physiological variation could be the source of their gut microbiota differences; in turn, the gut microbiota could involve and affect host physiology via multiple pathways, and then enhance personality phenotypes via these physiological changes, such as cerebral epigenetic marks [64], and the butyrate-producing [62].

Moreover, PICRUSt2-predicted KEGG pathway analysis showed that the bacterial functional pathways, which were related to porphyrin and chlorophyll metabolisms, were significantly enriched in bold gerbils. In contrast, the functional pathways involved in taurine, hypotaurine and tryptophan metabolism were significantly decreased. This result suggests that increased porphyrin and chlorophyll metabolisms and decreased taurine, hypotaurine and tryptophan metabolisms may be related to the development of individual boldness. Recent studies have reported that tryptophan metabolism could be linked to behavior and cognition [65], and taurine supplement could reinforce the blue tit (*Cyanistes caeruleus*) nestlings’ risk-taking tendency and be related to spatial learning performance [66]. Thus, an alteration in metabolism in the gut microbiota may be related to host personality traits such as boldness.

In conclusion, our present study found a potential relationship between gut microbiota and the boldness personality in Mongolian gerbils. Even though we could not find evidence that changes in gut microbiota can significantly change individual’s boldness score, the “bold microbiota” reinforced the gerbil’s bold behavior obviously. Based on our current findings and the mixed results of other studies on personalities and gut microbiota [1,22,23,48], it would appear that the presence and direction of the relationship between individual personalities and gut microbiota are complex. There is thus a need for additional empirical and theoretical studies to further illuminate generalities about the nature of the relationship between animal personality and gut microbiota.

## Figures and Tables

**Figure 1 microorganisms-10-01054-f001:**
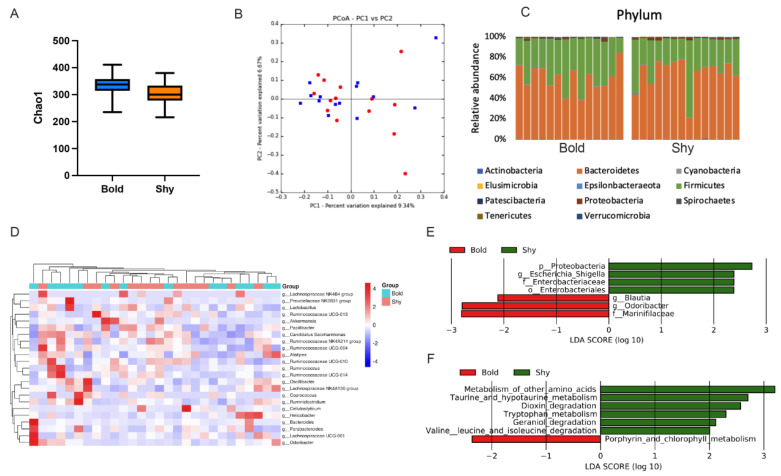
The comparisons of gut microbiota diversity between bold and shy personalities in Mongolian gerbils. (**A**) Alpha diversity (Faith’s phylogenetic diversity) of bacterial communities across groups. (**B**) PCoA plot based on Bray–Curtis distance metrics representing the differences in fecal microbial community structure in different groups (PERMANOVA). (**C**) Abundance represented as the proportions of ASVs classified at the phylum rank. (**D**) Cluster heatmap showing the proportions of ASVs classified at the genus rank. (**E**) Differential bacterial taxonomy selected by LEfSe analysis with LDA score > 2 in microbiota. (**F**) Predicted microbial functions using PICRUSt. Relative abundances of KEGG pathways by LEfSe analysis with LDA score > 2. Data are means ± SEM.

**Figure 2 microorganisms-10-01054-f002:**
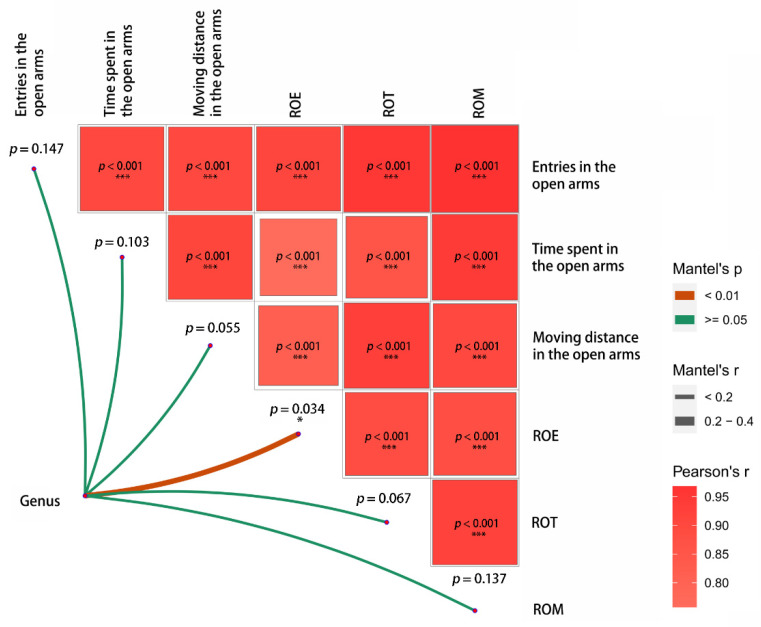
Pairwise comparisons of boldness aspects with a color gradient denoting Pearson’s correlation coefficient. ROE: ratios of open to closed arms entries; ROT: ratios of open to closed arms time spent; ROM: ratios of open to closed arms moving distance. Taxonomic of genus was related to each boldness indexes by Mantel tests. Edge width denotes the Mantel’s r value for the corresponding distance correlations, and edge color indicates the statistical significance. * *p* < 0.05, *** *p* < 0.001.

**Figure 3 microorganisms-10-01054-f003:**
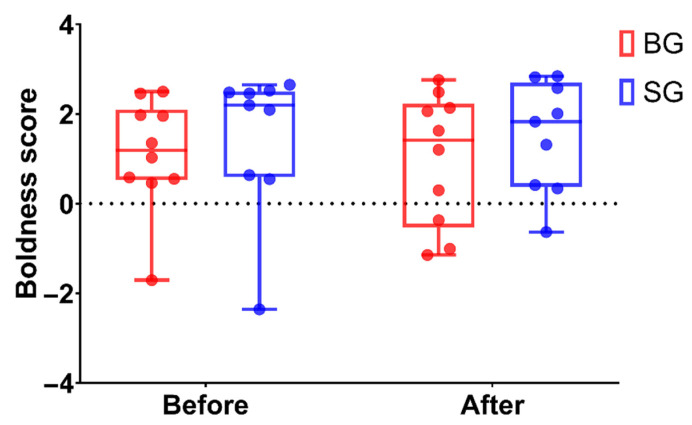
The boldness score of Mongolian gerbils before and after gut microbiota transplantation. Data are means ± SEM. (BG: “bold fecal” gavage group, SG: “shy fecal” gavage group.).

**Figure 4 microorganisms-10-01054-f004:**
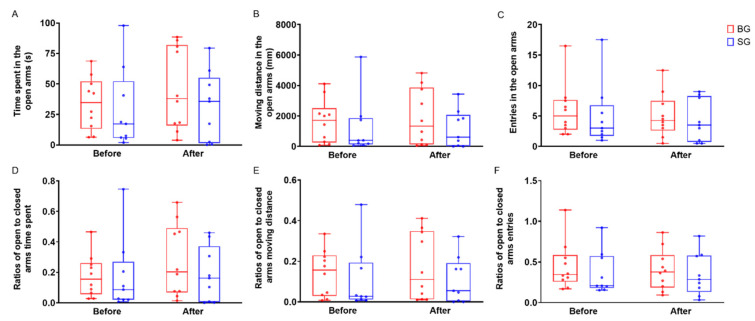
Boldness behavior variables change in Mongolian gerbils before and after fecal microbiota transplantation. (**A**) time spent in the open arms; (**B**) moving distance in the open arms; (**C**) entries in the open arms; (**D**) ratios of open to closed arms entries; (**E**) ratios of open to closed arms time spent; (**F**) ratios of open to closed arms moving distance. Data are means ± SEM. (BG: “bold fecal” gavage group, SG: “fecal” gavage group).

**Table 1 microorganisms-10-01054-t001:** Eigenvalues and eigenvectors of the first two components (PC1 and PC2), showing the percentage of variance.

Parameters	Component 1 (PC1)	Component 2 (PC2)
Entries in the open arms	−0.4207	0.2240
Time spent in the open arms	−0.4160	−0.1926
Moving distance in the open arms	−0.3769	−0.7419
ROE	−0.4028	0.4844
ROT	−0.4203	0.3209
ROM	−0.4112	−0.1569
Eigenvalue	5.04	0.47
Total variance (%)	84.04%	7.83%

ROE: ratios of open to closed arms entries; ROT: ratios of open to closed arms time spent; ROM: ratios of open to closed arms moving distance.

## Data Availability

The data that support the findings of this study are available from the corresponding author upon reasonable request. The raw sequence data are available in the NCBI Sequence Read Archive under the accession number (PRJNA818461).

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
