# Peer review of "The Gut Microbiota May Affect Personality in Mongolian Gerbils"

_microorganisms, 2022, doi:10.3390/microorganisms10051054_

Round 1

Reviewer 1 Report

General comment:

The topic of this work is scientifically intriguing. The link between gut microbial community and host behavioural characteristics is one of the most interesting frontiers in microbiome research and the idea behind this paper is nice, but I found the paper had some major problems. First, the language. The English needs a deep work of editing, because the paper is full of meaningless sentences and it is very difficult to read. Moreover, the study design is not clearly described (e.g., the real number of subjects involved is not so clear) and no detail about the outcome of the sequencing procedure is given in the manuscript. Details about the reads obtained, the reads that were discarded during data pre-processing, the rarefaction level that was adopted are only some of the information that should be included when describing a microbiome study. Lastly, the results are not described in detail and they present very rapidly the outcomes of the research.

Detailed comments:

  • Line 14: missing “the”
  • Line 21; phyla (plural)
  • Line 38: showing
  • Line 42: I would not use “to perform” in this sentence
  • Line 43: “we have not fully understood”, maybe?
  • Line 48: revealed?
  • Line 48: immune system
  • Line 49: I would drop the comma after “which”
  • Lines 72-74: please revise the English, the questions are not grammatically correct
  • Line 86: missing whitespace between “to” and “12”
  • Line 86: From here, it seems 137 gerbils were available, but only 48 (14+14+20) were involved in the study design: is it correct? If so, why did you drop the numbers so drastically? How did you select the gerbils among all the available ones in the three groups (bold, shy and middle)? I think such a strong cut in the number of subjects needs some other words to give the reader the opportunity to understand why you planned to proceed like you did.
  • Lines 86-87: something is really wrong with the grammar in this sentence. Please rephrase it.
  • Lines 90-92: also this sentence needs to be revised, for example: “one group, called “bold fecal” gavage (BG) group, received bold gerbils gut microbiota”
  • Line 95: were housed?
  • Lines: 107-108: check the font
  • Line 108: where used in
  • Line 108: principal, not principle
  • Line 110: I cannot link the numbers. If the gerbils were 137, one group has to be formed by 45, not 46 gerbils.
  • Line 122: Why did you use QIIME1 instead of QIIME2? I think a paper that aims to be published in 2022 should not be based on a pipeline that was discontinued since January 1st, 2018. Many important changes have been made from the 1.9.1 version, e.g., the switch from OTUs to ASVs. These changes truly affect many central aspects in a microbiota study like the one presented in this paper, such as the richness estimation or the accuracy of taxonomic assignment.
  • Line 123: you speak about metrics (plural), but you only cite one (Faith’s PD). Please add also the other one you included in this study.
  • Line 141: to analyze
  • Line 142: to compare (or, better, to test)
  • Line 143: were different?
  • Line 168: Faith’s PD values included in the plot seem to be 10 times higher than the common values calculated on OTUs you can find in other papers.
  • Line 169: please revise the English
  • Line 175: phyla
  • Line 175: what do you mean when you write “Firmicutes / Bacteroidetes”? Are you referring to the ratio?
  • Lines 179-180: significantly higher?
  • Line 206: capital letter missing at the beginning
  • Line 212: significantly
  • Line 212: remained
  • Lines 235-236: this sentence is really unclear
  • Line 237. Should influence
  • Line 239: I would not use “to perform” here
  • Line 241: what is the meaning of this sentence?
  • Line 244: Furthermore?
  • Line 251: Proteobacteria and Firmicutes are not genera, but phyla
  • Line 254: Consistently?
  • Line 263: please revise the English
  • Lines 265-268: this sentence is really unclear

Reviewer 2 Report

Dear Authors,

The manuscript entitled "Gut microbiota may affect the personality of Mongolian gerbils" addresses the issue with appropriate experiments and interesting results. However, I have some comments.

  1. In introduction-Authors should mention how bidirectional communication is (vagus nerve, immune system, systemic circulation).
  2. Line 70-Clarify whether it is the gut microbiota or the composition of the gut microbiota, i.e. the balance between strains.
  3. Were only male gerbils used? If so, what is the reason for not using female gerbils?
  4. In discussion-The authors should hypothesize some possible pathway to explain how the FMT reinforces the accepted gerbil's bold behaviour phenotypes.
  5. Line 159- Separate thePC1

Author Response

Response to reviewers

We gratefully thank the editor and all reviewers for their time spend making their constructive remarks and useful suggestions, which has significantly raised the quality of the manuscript and has enable us to improve the manuscript. Each suggested revision and comment, brought forward by the reviewers was accurately incorporated and considered. Below the comments of the reviewers are response point by point and the revisions are indicated.

Reviewer 2

Comments and Suggestions for Authors

The manuscript entitled "Gut microbiota may affect the personality of Mongolian gerbils" addresses the issue with appropriate experiments and interesting results. However, I have some comments.

  1. In introduction-Authors should mention how bidirectional communication is (vagus nerve, immune system, systemic circulation).

Answer: Thanks for your constructive suggestion. We have added some examples to describe the bidirectional communication between gut bacteria and the brain. Please see lines 36-37 of the revised manuscript

  1. Line 70-Clarify whether it is the gut microbiota or the composition of the gut microbiota, i.e. the balance between strains.

Answer: Thank you for your comment. We have modified the sentence according to your suggestion. Please see lines 71 of the revised manuscript

  1. Were only male gerbils used? If so, what is the reason for not using female gerbils?

Answer: Thanks for your comment. Yes, we only use the male gerbils, because of the physiological state of female gerbils are relatively unstable, thus, in this study we only used the male gerbils.

  1. In discussion-The authors should hypothesize some possible pathway to explain how the FMT reinforces the accepted gerbil's bold behaviour phenotypes.

Answer: Thanks for your constructive suggestion. We have modified our manuscript and added some discussion of the possible pathway to explain how the FMT reinforces the accepted gerbil's bold behaviour phenotypes. Please see lines 318-322 of the revised manuscript

  1. Line 159- Separate thePC1

Answer: Thank you for your suggestion. In table 1, we show the first two components are want to explained why we use PC1 as the boldness score, and we have modified our text in results and wish to make it clearer.

Round 2

Reviewer 1 Report

I think the authors addressed almost all the points very carefully. I think a language revision is still needed (I detected some more English errors here and there), but the paper has been significantly improved.

I just have a main question about the new outputs. Indeed, the fact that some results do not change even slightly (Figure 1E, Figure 2) leaves me somewhat perplexed. The effect of changing from QIIME1 to QIIME2 deeply affects the count table at the lowest level (ASV), as proved, for example, by the great changes in Faith PD values. I think this effect should be seen also at genus level, even if in a considerably lower measure. Could it be possible that the old version of these figures has been uploaded in place of the new one? 

Author Response

Response to reviewers

We gratefully thank the editor and all reviewers for their time spend making their constructive remarks and useful suggestions, which has significantly raised the quality of the manuscript and has enable us to improve the manuscript. Each suggested revision and comment, brought forward by the reviewers was accurately incorporated and considered. Below the comments of the reviewers are response point by point and the revisions are indicated.

Reviewer 1

General comment:

The topic of this work is scientifically intriguing. The link between gut microbial community and host behavioural characteristics is one of the most interesting frontiers in microbiome research and the idea behind this paper is nice, but I found the paper had some major problems. First, the language. The English needs a deep work of editing, because the paper is full of meaningless sentences and it is very difficult to read. Moreover, the study design is not clearly described (e.g., the real number of subjects involved is not so clear) and no detail about the outcome of the sequencing procedure is given in the manuscript. Details about the reads obtained, the reads that were discarded during data pre-processing, the rarefaction level that was adopted are only some of the information that should be included when describing a microbiome study. Lastly, the results are not described in detail and they present very rapidly the outcomes of the research.

Answer: Thanks for your constructive suggestion, which is highly appreciated. We have carefully scrutinized the manuscript, and made corresponding revisions including some description, typos, grammatical errors and long sentences, etc. We modified our analysis methods according to reviewer’s suggestion and modified our results and discussions.

Detailed comments:

Line 14: missing “the”

Answer: Thank you for your comments. We have modified the sentence. Please see lines 14-15 of the revised manuscript.

Line 21; phyla (plural)

Answer: Thank you for your comments. We have modified the word “phylum” to “phyla”. Please see lines 21 of the revised manuscript.

Line 38: showing

Answer: Thank you for your comments. We have modified the word according to the comment. Please see lines 40 of the revised manuscript.

Line 42: I would not use “to perform” in this sentence

Answer: Thanks for your kind suggestions, we have modified the sentence. Please see lines 44 of the revised manuscript.

Line 43: “we have not fully understood”, maybe?

Answer: Thank you for the suggestion. We have modified this expression throughout the text according to the comment. Please see lines 45 of the revised manuscript.

Line 48: revealed?

Answer: Thank you for your comments. We have modified the word according to the comment. Please see lines 49 of the revised manuscript.

Line 48: immune system

Thank you for your comments. We have modified the word according to the comment. Please see lines 49 of the revised manuscript.

Line 49: I would drop the comma after “which”

Answer: Thank you for your comments. We have modified the word according to the comment. Please see lines 50 of the revised manuscript.

Lines 72-74: please revise the English, the questions are not grammatically correct

Answer: Thank you for your comments. We have modified the sentence according to the comment. Please see lines 73-75 of the revised manuscript.

Line 86: missing whitespace between “to” and “12”

Answer: Thank you for your comments. We have added whitespace between “to” and “12”. Please see lines 87 of the revised manuscript.

Line 86: From here, it seems 137 gerbils were available, but only 48 (14+14+20) were involved in the study design: is it correct? If so, why did you drop the numbers so drastically? How did you select the gerbils among all the available ones in the three groups (bold, shy and middle)? I think such a strong cut in the number of subjects needs some other words to give the reader the opportunity to understand why you planned to proceed like you did.

Answer: Thank you for the suggestion. We measured all of our gerbils’ boldness at 8 to 12 months. In order to make our data more objective, we choose the highest 14 and lowest 14 individuals as bold and shy gerbils to analysis gut microbiota, and choose the 20 middle scores of gerbils to FTM. Please see lines 87-93 of the revised manuscript.

Lines 86-87: something is really wrong with the grammar in this sentence. Please rephrase it.

Answer: We appreciate for your valuable comment. We have revised the text to address your concerns and hope that it is now clearer. Please see lines 87-93 of the revised manuscript.

Lines 90-92: also this sentence needs to be revised, for example: “one group, called “bold fecal” gavage (BG) group, received bold gerbils gut microbiota”

Answer: Thank you for the suggestion. We have modified this expression throughout the text according to the comment. Please see lines 91-93 of the revised manuscript.

Line 95: were housed?

Answer: Thank you for your comments. We have added the housing condition in our methods. Please see lines 93-94 of the revised manuscript.

Lines: 107-108: check the font

Answer: Thank you for your comments. We have check the font and modified it.

Line 108: where used in

Answer: Thank you for your comments. We have modified the sentence according to the comment. Please see lines 116 of the revised manuscript.

Line 108: principal, not principle

Answer: Thank you for your comments. We have modified the word according to the comment. Please see lines 116 of the revised manuscript.

Line 110: I cannot link the numbers. If the gerbils were 137, one group has to be formed

by 45, not 46 gerbils.

Answer: Thank you for your comments. We have checked our data and modified our description.

Line 122: Why did you use QIIME1 instead of QIIME2? I think a paper that aims to be

published in 2022 should not be based on a pipeline that was discontinued since January 1st, 2018. Many important changes have been made from the 1.9.1 version, e.g., the switch from OTUs to ASVs. These changes truly affect many central aspects in a microbiota study like the one presented in this paper, such as the richness estimation or the accuracy of taxonomic assignment.

Answer: Thanks for your constructive suggestion, which is highly appreciated. We have re-analysis our data use QIIME2, and modified this part throughout the text according to the comment. Please see lines 125-154 of the revised manuscript.

Line 123: you speak about metrics (plural), but you only cite one (Faith’s PD). Please add also the other one you included in this study.

Answer: Thanks for your constructive suggestion. We are not only made Faith’s PD, and added the Chao1 and Shannon diversity in our manuscript. Please see Fig 1A and Fig S1. Line 141: to analyze

Answer: Thank you for your comments. We have modified the sentence according to the comment. Please see lines 170 of the revised manuscript.

Line 142: to compare (or, better, to test)

Answer: Thank you for your comments. We have modified the sentence according to the comment. Please see lines 171 of the revised manuscript.

Line 143: were different?

Answer: Thank you for your comments. We have modified the sentence according to the comment. Please see lines 172 of the revised manuscript.

Line 168: Faith’s PD values included in the plot seem to be 10 times higher than the common values calculated on OTUs you can find in other papers.

Answer: Thank you for your comments. We corrected the results via the new analysis. Please see lines 199-220 of the revised manuscript.

Line 169: please revise the English

Answer: We appreciate for your valuable comment. We have revised the text to address your concerns and hope that it is now clearer. Please see lines 199-200 of the revised manuscript.

Line 175: phyla

Answer: Thank you for your comments. We have modified the word according to the comment. Please see lines 208 of the revised manuscript.

Line 175: what do you mean when you write “Firmicutes / Bacteroidetes”? Are you

referring to the ratio?

Answer: Thank you for your comments. “Firmicutes / Bacteroidetes” means “Firmicutes and Bacteroidetes”, we have modified our description to make this sentence clearer. Please see lines 208 of the revised manuscript.

Lines 179-180: significantly higher?

Answer: Thank you for your comments. Yes, from our results of LefSe, the p value was < 0.05.

Line 206: capital letter missing at the beginning

Answer: Thank you for your comments. We have modified the capital letter according to the comment. Please see lines 243 of the revised manuscript.

Line 212: significantly

Answer: Thank you for your comments. We have modified the word according to the comment. Please see lines 248 of the revised manuscript.

Line 212: remained

Answer: Thank you for your comments. We have modified the word according to the comment. Please see lines 248 of the revised manuscript.

Lines 235-236: this sentence is really unclear

Answer: We appreciate for your valuable comment. We have revised the text to address your concerns and hope that it is now clearer. Please see lines 269-272 of the revised manuscript.

Line 237. Should influence

Answer: Thanks for your kind suggestions, we have modified the sentence according to the comment. Please see lines 277 of the revised manuscript

Line 239: I would not use “to perform” here

Answer: Thanks for your kind suggestions, we have modified the sentence according to the comment.

Line 241: what is the meaning of this sentence?

Answer: Thank you for your comments. This sentence means the gut microbiota could influence the host physiological state which may alter the host behavior. We have rewrite this part and hope to make our manuscript clearer.

Line 244: Furthermore?

Answer: Thank you for your comments. We have modified the word according to the comment. Please see lines 281 of the revised manuscript.

Line 251: Proteobacteria and Firmicutes are not genera, but phyla

Answer: Thank you for your comments. We have modified the word “genera” to “phyla/genus”. Please see lines 285 of the revised manuscript.

Line 254: Consistently?

Answer: Thank you for your comments. We have modified the sentence according to the comment. Please see lines 290 of the revised manuscript

Line 263: please revise the English

Answer: We appreciate for your valuable comment. We have revised the text to address your concerns and hope that it is now clearer. Please see lines 302-304 of the revised manuscript

Lines 265-268: this sentence is really unclear

Answer: We appreciate for your valuable comment. We have revised the text to address your concerns and hope that it is now clearer. Please see lines 304-310 of the revised manuscript
